# Affective communication with patients with limited health literacy in the palliative phase of COPD or lung cancer: Analysis of video-recorded consultations in outpatient care

**Rebecca Otte** [1,2]*, **Ruud Roodbeen**[3], **Gudule Boland**[4], **Janneke Noordman**[1], **Sandra van Dulmen**[1,5]

**1** Nivel (Netherlands Institute for Health Services Research), Utrecht, the Netherlands, **2** Strategic Communication Group, Wageningen University and Research, Wageningen, the Netherlands, **3** Breuer&Intraval, Research and Consultancy, Research Department, Groningen, the Netherlands, **4** Pharos, Dutch Centre of Expertise on Health Disparities, Utrecht, the Netherlands, **5** Department of Primary and Community Care, Radboud University Medical Center, Radboud Institute for Health Sciences, Nijmegen, the Netherlands

* rebecca.otte@wur.nl

**Data Availability Statement:** Data from this study are available upon request due to the fact that the data contain potentially identifying or sensitive

## Abstract

### Objectives

Affective communication in outpatient care is important, especially in the palliative phase. Appropriate responses by healthcare providers to emotional cues or concerns let patients express their feelings and enhance information recall and patient satisfaction. Patients with limited health literacy experience more barriers in health-related communication and information, which makes recognizing their cues and concerns even more relevant. This study explores emotional cues/concerns expressed by patients with limited health literacy and evaluates healthcare providers' responses to these utterances.

### Methods

Verona Coding Definitions of Emotional Sequences (VR-CoDES), a consensus-based system for coding patients' expression of emotional distress in medical consultations, was used in this exploratory observational study to analyse affective communication in video-recorded outpatient consultations. Consultations of 18 (10 female, 8 male) COPD or lung cancer patients (aged 70.3±6.8) with limited health literacy were recorded and analysed. Eight healthcare providers in four hospitals participated in the study.

### Results

101 cues and 11 concerns were observed, making 6.2 (SD = 4.2) cues or concerns per consultation. Healthcare provider responses were explicit in 56% and left scope for further disclosures in 58% of the cases. Patients with limited health literacy seem to express more cues or concerns than other patient populations. Healthcare providers responded roughly equally often in five different ways, but they shied away from further exploring the emotion

patient information, and because they are owned by a third party, Nivel. The authors did not have any special privileged access to the data Requests for the data can be made to Nivel: Postal address: P.O. Box 1568 3500 BN Utrecht The Netherlands Tel: +31 (0)30 – 2729700 E-mail: receptie@nivel.nl Visiting adress: Otterstraat 118-124 3513CR Utrecht The Netherlands.

**Funding:** This study was funded by a grant from the Netherlands Organization for Health Research and Development, Palliative Care Programme (ZonMw, Palliantie: 844001403) (https://www.zonmw.nl/nl/). The funder had no role in study design, data collection and analysis, decision to publish, or preparation of the manuscript.

**Competing interests:** The authors have declared that no competing interests exist.

disclosed in the cue/concern. Future research should elaborate on these exploratory observations, especially regarding why HCPs often only leave limited space for further disclosure of emotions in palliative care.

## Introduction

Recognizing emotional distress in patients and anticipating emotional cues and concerns through affective communication is very important in clinical practice. This is even more so in the palliative phase, where the sometimes rapid and life-determining disease progression can raise existential questions and intense emotions [1]. Addressing patients' emotional responses by healthcare providers (HCPs) being empathic, reassuring and supportive is called 'affective communication' [2]. During affective communication, HCPs should listen to patients' concerns, provide comfort and encourage the physician-patient relationship [3]. According to Neumann et al. [4] and supported by the studies of Jansen et al. [5] and Westendorp et al. [6], affective communication helps patients to discuss their concerns with their HCP and helps them make better healthcare decisions through enhancing information recall.

Patients with limited health literacy may face additional challenges as they are, in general, less familiar with medical concepts and professional jargon and ask fewer questions during physician-patient interactions [7]. Health literacy is defined as 'the knowledge, motivation and competences to access, understand, appraise, and apply health information to make judgments and take decisions in everyday life concerning healthcare, disease prevention and health promotion to maintain or improve quality of life throughout the course of life' [8]. In 2015, almost 48% of the European population had limited health literacy [9] and in 2019 it was estimated that 28.9% of the adult Dutch population had limited health literacy skills [10]. Limited health literacy is predominantly observed in people with less than ten years of formal education and is also associated with lower socioeconomic status (SES) [11]; effectively meeting the emotional needs of these patients is therefore also a matter of health equity. Patients with limited health literacy often feel ashamed about not understanding information and this stops them asking for clarification and prevents active participation in the decision-making process [12, 13].

This study was conducted among patients with advanced lung cancer or COPD (chronic obstructive pulmonary disease), a population known for having a high prevalence of limited health literacy [12] and high rates of lifelong smoking [13–15]. The prevalence of COPD is higher among people with primary school as their highest level of education (13%) compared to people with a higher level of education (2%). Furthermore, COPD is more prevalent in people with low incomes (7%), compared to the highest income group (3%) [16]. This makes this patient population particularly suitable and relevant for investigating limited health literacy.

Until now, research on affective communication with cancer patients focused predominantly on the curative phase, without considering health literacy [17, 18]. Indeed, the patient population with advanced lung cancer and/or COPD is often excluded from research in the palliative phase [19]. This study addresses this gap by investigating the affective communication in the palliative phase between lung patients with limited health literacy and their HCPs. This study uses a validated approach using the Verona Coding Definitions of Emotional Sequences (VR-CoDES) [20], which is well-suited for assessing the emotional aspects of patient-provider communication. Researchers may use VR-CoDES-CC to label both explicit and subtle expressions (concerns and cues, respectively) of emotional distress [21].

Recognizing and exploring such emotional cues and concerns is important because it lets patients express their feelings, enhances information recall and ultimately results in improved patient satisfaction and facilitates information recall in patients [5].

## Methods

### Design

This cross-sectional observational study involved video recordings of lung patient consultations in outpatient care in a project called 'A basic understanding' ('Goed Begrepen.' in Dutch). This project represents a multicentre study that aims to improve the provision of information for and shared decision-making with patients with limited health literacy in the palliative phase in outpatient care. Video recordings of consultations were made between April and October 2018 at the oncology and pulmonology departments of four Dutch hospitals (three university hospitals and one general hospital). Before participation, written informed consent of both the HCP and the patient was obtained. If necessary, the research team was available to explain the informed consent to the patient and their family members. Only the HCP was visible in video recordings of the consultations. The study protocol was evaluated by the CMO Regio Arnhem/Nijmegen, who stated that the study did not need formal ethical approval (CMO file number: 2017–3623). All participants were informed (verbally and in writing) about the study and signed an informed consent form. Participants' anonymity was assured and they could withdraw from the study at any time.

### Patient sample

In this study, the WHO definition of palliative care was applied [22], referring to patients facing problems associated with life-threating illness, in which 'palliative' might refer to a disease that is not amenable to a curative approach. Patients were included if they were aged ≥ 18, diagnosed with lung cancer or chronic obstructive pulmonary disease (COPD) and in the palliative phase of their disease assessed by the participating medical professional or local project manager with a medical background, capable of making this assessment. Furthermore patients were only included if professionals considered them to be LHL (expert opinion), if their educational level was secondary vocational education at most ('lower-level' education according to the definition of Statistics Netherlands [23]) and/or if they had LHL. LHL was assumed when patients answered affirmatively to at least one of three screening questions: *"Many people experience difficulties reading hospital leaflets. How about you*?", *"Many people find forms and filling them in difficult. How about you*?" and *'Do you need help filling in forms or reading leaflets*?". These are based on the single-item question of Chew et al. [24] The expert opinion prevailed the educational assessment and screening questions, as research has shown that LHL is not only found in people with a lower educational level. Moreover, out of embarrassment or shame, patients often hide the fact that they have trouble understanding health related information [25]. Patients were excluded from participating if they 1) were in the terminal phase of their disease (only patients with COPD and lung cancer with and without metastases were included in the palliative phase of their disease), 2) had a severe intellectual disability, or 3) had a psychiatric problem or dementia. The professionals included were physicians and nurses who regularly conduct consultations with patients with cancer or/and COPD and discuss palliative care and/or treatment options. Consultations in which the HCP primarily discussed the situation of the patient with another person (e.g. a partner or child) were excluded from the analysis.

## VR-CoDES-CC

The Verona coding definitions of emotional sequences (VR-CoDES) provides a method for analysing affective communication during medical visits. First, VR-CoDES-CC was used to assess affective communication by coding concerns and cues expressed by the patients [26]. Cues are defined as 'verbal or nonverbal hints that suggest an underlying unpleasant emotion'. These cues needed further clarification by the HCP. A concern is defined as 'a clear and unambiguous expression of an unpleasant current or recent emotion', i.e. when a patient explicitly verbalizes an emotion. Cues and concerns can be categorized as either healthcare provider elicited (HPE) or patient-elicited (PE). A cue or concern is defined as HPE if the expression of that cue or concern is 'solicited, explored or facilitated by the healthcare provider'. These cues/ concerns are 'coherently and logically connected with the healthcare provider's previous turn' and are an indicator of the space given to patients to explicate their concerns [20]. PE cues or concerns are introduced by the patient 'without having been solicited, invited or expected by the HCP to do so' [20]. See Table 1 for further definitions and examples.

## VR-CoDES-P

VR-CoDES-P [21] were used to code the response of the HCP to the cues or concerns of the patient. Similarly to the distinction between a cue and concern being based on the explicitness of the emotional component of the concern, this same is considered important in the response of the HCP [21]. If the HCP responded with an unambiguously specific or explicit reference to the words in the preceding cue or concern of the patient, the response was coded as 'explicit'. If the HCP's response did not specifically or explicitly mention the content or the emotion of the cue or concern, the response was coded as 'non-explicit'. Furthermore, after coding the responses as explicit or non-explicit, the coder analysed the response to see whether the response increased or reduced the space for further disclosure. According to Del Piccolo et al., [21] providing space referred to any response that actively or passively invites or allows the patient to say more about their cue or concern. The response is coded as reducing space if the HCP reduces scope for further disclosure by e.g. ignoring the cue or concern, switching subject, or postponing or actively blocking a discussion about the issue. In total, the responses of the HCP were labelled using five different codes (1. non-explicitly reducing space, 2. non-explicitly providing space, 3. explicitly reducing space, 4. explicitly providing space for the content and 5. explicitly providing space for the emotion within the cue or concern). Examples with quotes from the study population are given in Table 1.

## Analysis

The main coder (RO) was trained in VR-CoDES-P and VR-CoDES-CC using training material provided on the EACH website (www.each.eu). All videos were coded using the Behavioral Observation Research Interactive Software (BORIS) [27].

## Reliability of VR-CoDES-P and VR-CoDES-CC

The intra-class correlation coefficient (ICC) [28] was used to assess the inter-rater reliability of VR-CoDES-P and VR-CoDES-CC. The overall ICC for detecting the average cues and concerns in all four consultations was 0.938. The cues and concerns of the second coder were similar to those of the first coder in 78% of cases. Subsequently, the four consultations were discussed and adapted if needed. The codes for the HCP's responses for the second coder matched those of the first coder in 64% of cases for explicit versus non-explicit and 86% for providing versus reducing space in the total set of video consultations.

**Table 1. Description of the expressions or verbalization of the cues or concerns of the patients and the responses of the HCP according to Zimmermann et al. [20] and Del Piccolo et al. [21] with examples from the video recorded consultations.**

| Subject | Variable | Description |
|---|---|---|
| Patient | Cue | Verbal or non-verbal hints that suggest an underlying unpleasant emotion. Instances include vague or unspecified words to describe emotions, verbal hints to hidden concerns, words or phrases that emphasize physiological or cognitive correlations, neutral expressions that stand out and mention potential emotionally important issues (stressful life events and conditions), a patient-elicited repetition of a previous neutral expression, a non-verbal cue or a clear and unambiguous expression of an unpleasant emotion that is in the past or referred to in some unclear period of life. |
| | | *Example* |
| | | *Patient*: "Yesterday I said, because the help was there, that I'd be better off dead, because this is no kind of life anymore." |
| | Concern | A clear and unambiguous expression of an unpleasant current or recent emotion where the emotion is explicitly verbalized, with or without a stated issue of importance for the patient. |
| | | *Example* |
| | | *Patient*: "Yes, I'm scared because I don't know how long I have to live." |
| Healthcare provider | Explicit response | To be coded as explicit, a response should include either a specific or an explicit reference to the words in the preceding cue or concern, or be very clear in that it unambiguously refers to the stated cue or concern. |
| | | *Example (reducing space)* |
| | | *Patient* "I'm scared of hyperventilation" *HCP*: "Right, but if you're so short of breath, you will automatically hyperventilate". |
| | Non-explicit response | A non-explicit response is any response that does not specifically or explicitly mention either the content or the emotion of the cue or concern, or is ambiguous. The word non-explicit does not presume any recognition of the concern. |
| | | *Example (providing space)* |
| | | *Patient*: "No, when I first heard, it was, it was absolutely hellish." *HCP*: "Right. Right?" |
| | Response providing space | Providing space refers to any response that actively or passively invites or allows the patient to say more about the cue/concern or worry. This lets the healthcare provider give space for further disclosure. |
| | | *Example (explicit)* |
| | | *Patient (emotionally)*: "My brother-in-law last week and now it's my turn…" *HCP*: "Your brother-in-law passed away, or…?" |
| | Response reducing space | The response of the healthcare provider is rated as reducing space for further disclosure when the cue or concern is ignored, the subject is changed, discussion of the cue/concern is postponed, or when the opportunity for the patient to say more about the cue or concern is blocked or reduced. |
| | | *Example (non-explicit)* |
| | | *Patient*: "No but, um, the stress…" *HCP*: "Alright then, we are going to look right away." |

## Results

### Sociodemographic and clinical characteristics of the sample

In total, five practising pulmonologists, one pulmonologist in training and two specialized nurses participated in this study. Outpatient consultations with 18 patients were recorded. Nine patients (50%) were aged between 60 and 70, eight (44%) were aged between 71 and 80 and one was aged 81 or over (see Table 2). Twelve patients were accompanied by children or a partner; however, the cues and concerns of the accompanying people were not coded. The

**Table 2. Sociodemographic characteristics of the patient population, their diseases and the type of consultation.**

| | Total population ($n$ = 18) |
|---|---|
| Age (years) | 70.3 (± 6.8) |
| Sex (female) | 10 |
| Sex (male) | 8 |
| Low level of education | 16 |
| Medium level of education | 2 |
| COPD | 10 |
| Lung cancer | 8 |
| First consultation | 2 |
| Follow-up consultation | 16 |

consultations lasted between 6.25 minutes and 69.35 minutes with a mean of 21.51 minutes and standard deviation (SD) of 15.54 minutes. In total, 388 minutes of video-recorded consultations were observed. More information about the patient characteristics, their diseases and the type of consultation is presented in Table 2.

## Coding cues and concerns

Cues were found in 17 (94%) consultations and concerns in 6 (33%). Male and female patients expressed approximately the same number of cues and concerns (a mean of 6.4 cues and concerns per consultation for women and 6.0 for men). The length of the consultation was positively correlated with the total number of cues and concerns expressed by the patient (Pearson correlation coefficient 0.74). In our small sample, a little more than half of the concerns (55%) were initiated by the patient and coded as 'patient-elicited' (see Table 3 and Quote 1).

Quote 1

Patient: "I'm getting, I'm getting sick of it, really. Because every time it's something different again."

The majority of expressed cues were, however, 'healthcare provider elicited' (72%), and thus connected with the healthcare provider's previous expression (Quote 2).

Quote 2

HCP: "You also mentioned last time, that, the way things are going right now, it is not bearable anymore. Is that still the case?"

Patient: "That is still the case indeed"

## Coding HCP responses

Responses of the HCP to the emotional cues and concerns of the patients were divided into five categories (examples of these categories are presented in Table 1). Table 4 gives an overview of the various responses found in this study and the frequency with which the HCP used one of these five different categories of responses as a reaction to a cue or concern of a patient.

Of the 153 responses by the HCP to the cues and concerns of a patient, 61 were non-explicit; the HCP mentioned the content or the emotion of the cue or concern (either explicitly

**Table 3. Frequency, percentage, rate per minute and mean number of cues and concerns in all video-recorded consultations.**

| Variable | Frequency | Percentage (%) | Rate per minute | Mean number per consultation | Percentage patient-elicited (%) | Percentage healthcare provider elicited (%) |
|---|---|---|---|---|---|---|
| Cues | 101 | 90 | 0.26 | 5.6 | 28 | 72 |
| Concerns | 11 | 10 | 0.03 | 0.6 | 55 | 45 |
| **Total cues and concerns** | **112** | **100** | **0.29** | **6.2** | **31** | **69** |

**Table 4. Frequency and percentage of the responses (total 118) of the HCP on the cues and concerns of patients.**

| | Non-explicit response | Explicit response |
|---|---|---|
| *Providing space for content* | | *25* |
| *Providing space for emotion* | | *10* |
| **Total providing space** | **33** | **35** |
| **Total reducing space** | **28** | **22** |

or otherwise). Quote 3 presents an example in which the response of the HCP is coded as non-explicit, reducing space for further disclosure of the cue of the patient.

*Quote 3*

*Patient*: *"If it* (the hormone therapy, ed.) *doesn't work anymore*; *I don't want to suffer. I really don't want that."*

*HCP*: *"Right, right."*

The HCP could verbally or nonverbally provide space for further exploration of the cue or concern by a non-explicit response or an explicit response. Within the category 'explicit providing space', a distinction was made between explicitly providing space for the content or the affective, emotional side of the cue or concern. 35% of the responses of the HCP provider were coded as explicit responses, providing space for further disclosure. In 8% of the cases, the HCP referred to the emotion within the cue or concern (see quote 4) and in 21% of the cases to the content (see quote 5 and 6).

*Quote 4*

*HCP*: *"What do you enjoy most at the moment?"*

*Patient (emotionally)*: *"My children and my grandchildren* (crying). *And I don't want them to feel sad because of me."*

*HCP*: *"Sure* (nodding). *I understand that too, because it's something you get genuine enjoyment from, and it's what you see and will have to let go of sooner or later. That's also what you said at the beginning of the conversation?"*

*Quote 5*

*Patient*: *"I get tired so quickly, so quickly."*

*HCP*: *"Tired? Do you mean real fatigue, or is it puffing and being short of breath?"*

*Quote 6*

*Patient*: *"I feel very short of breath in the morning and in the evening. I don't know why that is either.(. . .) At such moments, I think I'm suffocating."*

*HCP*: *"Right, right. You know, or may not know, but I think that this is also discussed in Ciro* (treatment program specifically designed for patients with chronic lung conditions, e.d.), *which is that suffocation is a threatening idea for a lot of people. But if you have COPD, you don't die by suffocation. Suffocation actually means that there is no oxygen going into your lungs at al.'*

## Discussion

### Main findings

This study assessed the affective communication between lung patients with limited health literacy and their HCPs in the palliative phase of their disease. The mean number of cues and concerns expressed by patients with limited health literacy was 6.2 (SD = 4.2) per consultation, a rate of 1 cue or concern every 3.46 minutes. Furthermore, the five response categories (1. non-explicitly reducing space, 2. non-explicitly providing space, 3. explicitly reducing space, 4. explicitly providing space for the content and 5. explicitly providing space for the emotion within the cue or concern) were used approximately equally by HCPs except for number 5 (the explicit response, providing space for the affective, emotional aspect of the cue), which was only used in 8% of the cases.

### Added value of this study

**VR-CoDES-CC.** This study showed that emotional cues could be observed in almost all consultations, although concerns were present in only one third. This is consistent with other literature and shows that patients are more likely to express their feelings and worries indirectly, through cues, than directly via concerns [29, 30]. Most of the observed cues and concerns were HCP-elicited, while about half of the concerns were patient elicited. This is in contrast with the study by Del Piccolo et al. [29] in early-stage breast cancer patients, which found no differences in the number of cues/concerns which were HCP-elicited or patient-elicited, or the study of Finset et al. [30] in which the majority of the cues were initiated by the patient. However, neither Del Piccolo's nor Finset's study took patient health literacy into account. Because the majority of the cues observed in this study were responses to a previous remark by the HCP, this leads us to think that it might be possible that patients with limited health literacy are more likely to express their emotional cues after facilitation by the HCP, compared to patients in other studies and/or with other types of diseases. Moreover, as the majority of concerns were patient-elicited, this patient population may be more eager to express their concerns even when that is not directly facilitated by the HCP. Nevertheless, the relatively small number of observed concerns means that further research could expand our knowledge on this subject.

The mean number of observed cues and concerns per consultation in this study is almost twice as high as in the studies by Finset et al. [30] (with a mean of 3.0 (SD = 3.2)) and by Oguchi et al. [31] (with 3.33 (SD = 3.87) cues and 0.22 (SD = 0.624) concerns per consultation (33.3 min)). The study by Oguchi et al. [31] resulted in an average of one cue/concern every 10.1 minutes, which is more than twice as frequent as this study. However, neither Finset et al. [30] and Oguchi et al. [31] did specifically include patients in the palliative phase nor selected for health literacy. Only Amundsen et al. [32] did compare the curative and palliative phase using VR-CoDES and suggested that the setting (either curative or palliative) was not

significantly correlated to the amount of expressed cues or concerns in their study population. However, as there are several variables within this study design (e.g. setting, diagnosis and health literacy), comparison with existing literature is limited. To the best of our knowledge, there are no studies into affective communication in patients with limited health literacy in the palliative phase using VR-CoDES. Nevertheless, given that we know that patients with limited health literacy are more likely to experience difficulties in provider-patient communication [33, 34], it seems possible that these difficulties are reflected in the amount of expressed cues or concerns. In addition, the relatively high prevalence of cues and concerns about the fear of suffocation (Quote 6) would seem to be specific to the patient population (lung patients). Further studies, which take these difference settings and patient populations into account, will need to be undertaken.

**VR-CoDES-P.** The equal use of non-explicit and explicit responses in this study is not in line with the study by Zimmermann et al. [35] and the review by Finset et al., [36] which both suggest that HCPs' responses are more likely to be non-explicit, thereby reducing the space for further exploring the cue or concern. The reason for this is unclear, but might have something to do with the healthcare provider being aware of the limited health literacy level of the patient. This was emphasized by the healthcare provider asking questions such as "*Can you read leaflets*? *It sounds a bit weird, but can you do that*? *Some people can't and there are some things I find difficult to read too.*" This background information may have resulted in the response already being adapted and relatively more explicit to make it clearer for the patient. The ratio of responses reducing space (43%) and providing space (58%) is in line with Mjaaland et al., [37] who similarly found that physicians provide room for further disclosure in response to about half of all emotional cues and concerns and also reported a focus on the medical side (content) rather than the emotional side of the cue or concern.

**Methodological considerations.** The unique focus on palliative care patients with limited health literacy is one of the strengths of this paper. Although limited health literacy is a well-known concept in health communication, we are not aware of any research conducted on emotional cues and concerns of these patients and the responses of the HCP. This exploratory study therefore provides input for further research into affective communication with this patient population. However, there are also a few methodological considerations. This study can only be generalized to a reduced extent because of the limited number of consultations available for observation and the broad range of consultation durations (6 to 69 min). Furthermore, although video-recorded consultations provide more information than audio recordings, only the HCP was visible due to privacy restrictions. This may have resulted in underestimations of the (non-verbal) cues and concerns of the patient. Additionally, further distinctions could have been made in the responses, as described by Del Piccolo [21]; however, this observational study was limited to responses in the aforementioned categories (also described in Table 1) to obtain an indication of the general HCP responses.

## Conclusions

The results from this explorative study could help to indicate differences in affective communication between the general population and patients with limited health literacy. Notwithstanding the relatively limited sample, the findings suggest that patients with limited health literacy seem to express more cues or concerns than the general population. HCPs used five different kinds of responses approximately equally often but only explicitly provided space for further disclosure of the emotion of the cue/concern in a limited number of the cases, which is in line with other studies investigating HCP responses using VR-CoDES. HCPs tend to close themselves off from emotional communication, particularly if the emotion was clearly expressed in

the patient's cue or concern. This explorative study had a small sample size and future research should replicate our findings. Furthermore, future research could look into the reasons why HCPs in palliative care only provide space to a limited extent for further disclosure of the emotion. More research into the general population (those with sufficient health literacy) in the palliative phase would help create a better understanding of the differences in affective communication between patients with limited and sufficient health literacy.

## Acknowledgments

The authors would like to thank all the healthcare providers and patients who participated for their valuable contributions to the study. Furthermore, the assistance provided by Lotte Schulze as a second coder and the assistance of Wietske van der Hoeven in data collection was greatly appreciated. We also would like to thank Mike Wilkinson from Tessera Translations for the English language review.

## Author Contributions

**Conceptualization:** Rebecca Otte, Ruud Roodbeen, Gudule Boland, Janneke Noordman, Sandra van Dulmen.

**Investigation:** Rebecca Otte.

**Methodology:** Rebecca Otte.

**Supervision:** Janneke Noordman, Sandra van Dulmen.

**Validation:** Rebecca Otte.

**Writing – original draft:** Rebecca Otte.

**Writing – review & editing:** Rebecca Otte, Ruud Roodbeen, Gudule Boland, Janneke Noordman, Sandra van Dulmen.

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
