## [Decision Letter · Decision Letter 0]

23 Mar 2021

PONE-D-21-01177

Affective communication with patients with limited health literacy in the palliative phase of COPD or lung cancer: analysis of video-recorded consultations in outpatient care

PLOS ONE

Dear Dr. Otte,

Thank you for submitting your manuscript to PLOS ONE. After careful consideration, we feel that it has merit but does not fully meet PLOS ONE’s publication criteria as it currently stands. Therefore, we invite you to submit a revised version of the manuscript that addresses the points raised during the review process.

As you will see, both reviewers found your findings interesting, but they have concerns about the (small) sample. Hence, I would ask you to pay special attention in your revision explaining the exact patient sample in more detail and be more cautious in drawing conclusions based on the presented results. I also agree that additional qualitative analyses would strengten your manuscript considerably.

We look forward to receiving your revised manuscript.

Kind regards,

Barbara Schouten

Academic Editor

PLOS ONE

Journal Requirements:

3. Please provide more information on the consultations with Health care providers so that others could replicate the study. For example, if an interview guide was used in the study, please include a copy as Supporting Information.

Reviewers' comments:

Reviewer's Responses to Questions

**Comments to the Author**

1. Is the manuscript technically sound, and do the data support the conclusions?

Reviewer #1: Partly

Reviewer #2: Partly

2. Has the statistical analysis been performed appropriately and rigorously? 

Reviewer #1: N/A

Reviewer #2: Yes

3. Have the authors made all data underlying the findings in their manuscript fully available?

Reviewer #1: No

Reviewer #2: No

4. Is the manuscript presented in an intelligible fashion and written in standard English?

Reviewer #1: Yes

Reviewer #2: Yes

5. Review Comments to the Author

Reviewer #1: The present study describes expressions of emotional cues and concerns and the responses of physicians and nurses in a small sample of 18 consultations with COPD and lung cancer patients in palliative care. In the Introduction the authors point to the need to study the affective communication between patients with low health literacy in a palliative phased and their providers, which they describe as a gap in the literature. The aim of the study is to address this gap by investigating affective communication for the patient group.

The Results section is quite straightforward and descriptive, consisting mostly of counting the numbers and concerns of patients and of clinician responses, which are described as explicit vs. non-explicit in relation to cues/concerns and in terms of whether or not they provide space for further disclose of the cue and concerns.

In the Discussion, the authors make a point of the finding that while a majority of cues were clinician-elicited, concerns were more often patient-elicited. The sample is way too small draw conclusions from their data. If only three more concerns had been clinician elicited, the percentage of clinician elicited utterances had been the same as for cues.

The authors suggest that cues and concerns may be more prevalent in the palliative than in a curative phase of treatment and care and also that cues and concerns could be more prevalent among patients with limited health literacy. In evaluating the number of cues and concern per consultation, the authors compare their own findings with two other studies only, which both report lower frequencies. However, the literature includes a number of other studies applying VR-CoDES which report higher frequencies of cues and concerns. Moreover, the authors do not provide a comparison between low literacy and high literacy patients. The conclusion of the authors, that “patients with limited health literacy seem to express more cues and concerns than other patient population” cannot be drawn simply based on the reported findings.

To sum up, the study reports interesting descriptive data on affective communication with low literacy palliative patients, but the data set is not large enough to warrant the conclusion drawn in the Discussion section. A qualitative analysis of the data set might provide a more interesting picture of affective communication in consultations with this particular patient group.

Reviewer #2: Review PNOS affecitve communication, Verona Coding

General remarks:

The study presents interesting data about a very specific patient population: patients with advanced lung disease, either COPD or lung cancer with low health literacy. I could imagine that it was hard work to gather the data and appreciate the author’s attempt to have a closer look at some aspects of care for this vulnerable group of patients.

However, I have some major problems with the paper in its current form: I did not find a good definition of ‘palliative’ care. In this age group, the very diagnosis of COPD probably means that a curative option does not exist (lung transplantation?). In addition, lung carcinoma in general is not necessarily a palliative situation, even in this age group. Therefore, I would like to read who had assigned patients to the group of ‘palliative patients’ based upon which criteria? If ‘palliative’ is just the opposite of ‘curative’ one could assume that everybody in this age group with a chronic disease, be it coronary heart disease or chronic arthritis, is a palliative case because curative options are no longer available, etc.

The authors often compare their data to data from other patient populations, but I doubt whether this is justified. If neither patients in a palliative setting, as opposed to patients in early stages of breast cancer (Lidia paper) nor patients with low health literacy have been investigated, any difference found between data in this paper and the references cannot be attributed with certainty. Yet, I think as an observational study this paper is worth publishing, but I recommend being much more conservative in drawing conclusions based upon comparison with the existing literature.

In spite of my critical remarks I think these data are interesting because they show that low health literacy does not silence patients – they are still able to express their concerns (even in the form of patient-elicited concerns) and are willing and able to respond to hcp’s attempts to open space and bring in hints to negative underlying emotions. That is good news and should prompt hcp’s not to take a pessimistic stance like: “These patients don’t understand, what is going on”.

Some details are listed below:

Intro: Anticipating patients’ emotional responses by healthcare providers (HCPs) being empathic, reassuring and supportive is called ‘affective communication’

I’m not so sure whether ‘anticipating’ is the correct term here. The authors probably don’t assume that hcps possess the witch power to predict whether the next statement of a patient will be a concern? The very next sentence, listing characteristics of affective listening makes clear that this is not the case. Instead, the task here is not to ignore affective utterances but to respond to them with one of the quoted communication acts.

Methods:

‘Patients were also included if professionals considered them to be LHL (expert opinion).’

That’s an interesting idea, however, drawing from our experience with hcp’s inability to decide correctly, whether a patient needs the help of an interpreting service, the reader might wish to know how reliable such an estimation was. Why was this included in the inclusion criteria? How often did it occur?

‘patients were excluded from participating if they 1) were in the terminal phase of their disease’

This makes me think of the definition of a palliative phase. I understand that this does not necessarily refer to the final phase of a disease, but then, palliative might refer to a disease that is rather not amenable to a curative approach. Is that the definition used?

Discussion:

‘The cues and concerns of patients in the palliative phase with limited health literacy may be of a different order or more prevalent than in other patients in the curative phase.’

Besides a problem with phrasing (‘different order’?) I wonder whether this sentence is justified. We see a mixture of two factors mentioned: low versus high or average health literacy, and curative versus palliative phase of a disease. I think that these two factors cannot be disentangled within this design – there are no data on patients in a similar phase of their disease being non-low in health literacy.

‘This is relevant because the patients in this study expressed cues and concerns indicating distressing misunderstandings.’

Did I miss this point? Where do the authors present data on the nature of the cues and concerns that would indicate that they are related to misunderstandings? Is this based upon the assumption that high health literacy is effective against misunderstandings? Then, this connection should be made explicit. Basically, I am not so sure, whether this is the case. Couldn’t it be that hcps adjust their wording and the complexity of information to their perception of patients’ health literacy in such a way that ‘smarter’ patients are getting more information thereby offsetting the benefit of a higher health literacy?

The very next sentence in the discussion paper would rather strengthen my skeptical comment: patients refer to their fear of suffocation and that’s not a misunderstanding but harsh reality.

6. PLOS authors have the option to publish the peer review history of their article (what does this mean?). If published, this will include your full peer review and any attached files.

Reviewer #1: No

Reviewer #2: **Yes: **Wolf Langewitz

---

## [Author Response · Author response to Decision Letter 0]

2 Dec 2021

Reviewers comments (reviewer #1): 

In the Discussion, the authors make a point of the finding that while a majority of cues were clinician-elicited, concerns were more often patient-elicited. The sample is way too small draw conclusions from their data. If only three more concerns had been clinician elicited, the percentage of clinician elicited utterances had been the same as for cues. 

Thank you for your suggestions. We agree that the sample of this explorative study is small and these data must be interpreted with caution. This is also mentioned in the discussion, (‘This study can only be generalised to a limited extent because of the low number of consultations available for observation and the broad range of consultation durations’). However, when it comes to the formulation of the results, we indeed agree that some statements could be formulated with more caution. If applicable, we have used the word ‘might’. In this specific case, we think that emphasising the small study sample and presenting the percentages in the sentence itself also prevents misunderstanding. 

We have changed the sentence ‘The majority of the concerns were initiated by the patient and coded as ‘patient-elicited’ (see Table 3)’ to ‘In our small sample, a little more than half of the concerns (55%) were initiated by the patient and coded as ‘patient-elicited’ (see Table 3 and Quote 1).’ We also changed the wording in the Discussion part under ‘added value of this study’ from ‘the majority of the concerns’ to ‘about half of the concerns’. 

The authors suggest that cues and concerns may be more prevalent in the palliative than in a curative phase of treatment and care and also that cues and concerns could be more prevalent among patients with limited health literacy. In evaluating the number of cues and concern per consultation, the authors compare their own findings with two other studies only, which both report lower frequencies. However, the literature includes a number of other studies applying VR-CoDES which report higher frequencies of cues and concerns. Moreover, the authors do not provide a comparison between low literacy and high literacy patients. The conclusion of the authors, that “patients with limited health literacy seem to express more cues and concerns than other patient population” cannot be drawn simply based on the reported findings.

There are indeed two variables when it comes to comparing the findings with other studies; the palliative phase and health literacy of patients. The population of this study consists of lung cancer and COPD patients with limited health literacy in the palliative phase. Especially patients with limited health literacy is a specific population that is mostly not included in research. In the manuscript we have compared the results of our study to those of Del Piccolo et al.[1], Finset, Heyn and Ruland[2] and Oguchi et al.[3], which all included cancer patients in their study population. However, none of these studies did specifically include patients in the palliative phase nor selected for health literacy. To the best of our knowledge there are no studies into affective communication in patients with limited health literacy using the VR-CoDES. 

To address your remark to make a comparison between patients in the curative or palliative setting and between low and high literate patients, we have expanded the discussion by including the article of Amundsen et al.[4] in which it is shown that the setting (either curative or palliative) did not influence the number of cues and concerns expressed by patients at the Oncology Outpatient Clinic at the University Hospital of North Norway. Again, this study did not include LHL patients. This leads us to speculate that the increased number of cues and concerns expressed by this specific patient population is indeed related to the health literacy of the patients, rather than to the specific phase of the disease, either curative or palliative.

To sum up, the study reports interesting descriptive data on affective communication with low literacy palliative patients, but the data set is not large enough to warrant the conclusion drawn in the Discussion section. A qualitative analysis of the data set might provide a more interesting picture of affective communication in consultations with this particular patient group.

We agree that this explorative study warrants strong conclusions and have changed the wording of the results on several points to make clear that no strong conclusions can be made. Future research should replicate our findings. 

Thank you for the suggestion of a qualitative analysis of the data set. The main goal of this study was to give a broad overview of the most notable findings when it comes to affective communication in patients with limited health literacy. This is also the reason why we chose to distinguish only five categories in responses of the health care provider, to give a clear indication of the differences, within the limits of the data set of this explorative study. However we think that it is indeed a good idea to add more qualitative data to the manuscript and have added an additional number of quotes to underpin the findings of the study and provide a clearer picture of the expressed cues, concerns and responses. 

Reviewers comments (reviewer #2): 

The study presents interesting data about a very specific patient population: patients with advanced lung disease, either COPD or lung cancer with low health literacy. I could imagine that it was hard work to gather the data and appreciate the author’s attempt to have a closer look at some aspects of care for this vulnerable group of patients. However, I have some major problems with the paper in its current form: I did not find a good definition of ‘palliative’ care. In this age group, the very diagnosis of COPD probably means that a curative option does not exist (lung transplantation?). In addition, lung carcinoma in general is not necessarily a palliative situation, even in this age group. Therefore, I would like to read who had assigned patients to the group of ‘palliative patients’ based upon which criteria? If ‘palliative’ is just the opposite of ‘curative’ one could assume that everybody in this age group with a chronic disease, be it coronary heart disease or chronic arthritis, is a palliative case because curative options are no longer available, etc.

Thank you for your compliments and relevant critical question about the definition of ‘palliative’ care. In this study we have used the definition of ‘palliative care’ as provided by the WHO.[5] In earlier versions, the WHO definition of palliative care mainly stressed its relevance to patients not responsive to curative therapy, but in the latest version the WHO definition has been extended, including also other aspects (physical, emotional and spiritual needs of a patient).[6] When it comes to lung diseases the exact point of transition from a curative to a palliative approach is hard to define. This is also related to a subsequent comment, related to the methods. Indeed ‘palliative’ refers to a disease that is not amenable to a curative approach. We have adapted and extended the definition of ‘palliative care’ and ‘palliative’ in the methods. 

We agree with the reviewer that the explanation for the determination of patients being in the palliative phase of their disease needs further clarification. In this process, either the local project manager or health care provider of the participating hospital (specialized nurses or medical specialists, always professionals with a medical background capable of making this assessment) assessed whether patients were in the palliative phase of their disease for both cancer or COPD. In their assessment they adhered to the definition of palliative care provided in the adapted methods. 

In the manuscript, we have added information to the methods section, as presented below: 

‘In this study, the WHO definition of palliative care was applied[5], referring to patients facing problems associated with life-threating illness, in which ‘palliative’ might refer to a disease that is not amenable to a curative approach. Patients were included if they were aged ≥ 18 , diagnosed with lung cancer or chronic obstructive pulmonary disease (COPD) and in the palliative phase of their disease assessed by the participating medical professional or local project manager with a medical background, capable of making this assessment.’ 

The authors often compare their data to data from other patient populations, but I doubt whether this is justified. If neither patients in a palliative setting, as opposed to patients in early stages of breast cancer (Lidia paper) nor patients with low health literacy have been investigated, any difference found between data in this paper and the references cannot be attributed with certainty. Yet, I think as an observational study this paper is worth publishing, but I recommend being much more conservative in drawing conclusions based upon comparison with the existing literature.

We noticed that this comment is comparable with one of the feedback points of reviewer one. We have reviewed the discussion and conclusion in order to emphasize the exploratory design of this study and the corresponding limitations. Furthermore we agree that the comparison between the results of this study and existing literature can only be made to a limited extend, taking into account the different variables and settings (early stage breast cancer and the palliative phase). 

Intro: Anticipating patients’ emotional responses by healthcare providers (HCPs) being empathic, reassuring and supportive is called ‘affective communication’

I’m not so sure whether ‘anticipating’ is the correct term here. The authors probably don’t assume that hcps possess the witch power to predict whether the next statement of a patient will be a concern? The very next sentence, listing characteristics of affective listening makes clear that this is not the case. Instead, the task here is not to ignore affective utterances but to respond to them with one of the quoted communication acts.

You are indeed correct. We have changed ‘anticipating’ into ‘addressing’. 

Methods:

‘Patients were also included if professionals considered them to be LHL (expert opinion).’

That’s an interesting idea, however, drawing from our experience with hcp’s inability to decide correctly, whether a patient needs the help of an interpreting service, the reader might wish to know how reliable such an estimation was. Why was this included in the inclusion criteria? How often did it occur?

We agree with the reviewer that the explanation for the inclusion of health care provider – identified patient should be clarified. With regards applying only the expert opinion in assessing health literacy in patients, this was a rare occurrence. However, the opinion of the health care provider did always prevail the objective assessment of LHL using the screening questions and the patient’s educational level. There are several reasons why we choose to include the consideration provided by the health care provider. First of all patients often hide that they do not understand health information out of shame and could easily circumvent our assessment in the three screening questions.[7] Furthermore, LHL is not only found in people with a low educational level, but also in people with medium or high levels of education.

To clarify this in the manuscript we have changed this part of the methods section, indicating that the expert opinion prevailed the educational background and the three screening questions of LHL. 

 ‘patients were excluded from participating if they 1) were in the terminal phase of their disease’

This makes me think of the definition of a palliative phase. I understand that this does not necessarily refer to the final phase of a disease, but then, palliative might refer to a disease that is rather not amenable to a curative approach. Is that the definition used?

This is linked to a previous comment about the definition of ‘palliative’ care. Here you are indeed right and the definition of ‘palliative’ as being ‘not amenable to a curative approach’ is most suitable in this situation. We have adopted this suggestion in the description of the definition as added in the methods section and elaborated the explanation for the determination of patients being in the palliative phase of their disease. Whether the patients were in the palliative phase of their disease for both lung cancer and COPD was assessed by the local project manager or professional of the participating hospital. These were always professionals with a medical background capable of making this assessment (specialized nurses or medical specialists), and in their determination adhered to the definition of palliative care given in the manuscript. To clarify this in the manuscript, we have added information to the method section, as presented below: 

‘In this study, the WHO definition of palliative care was applied,[5] in which ‘palliative’ might refer to a disease that is not amenable to a curative approach. Patients were included if they were aged ≥ 18, diagnosed with lung cancer or chronic obstructive pulmonary disease (COPD) and in the palliative phase of their disease assessed by the participating medical professional or local project manager with a medical background, both capable of making this assessment.’ 

‘The cues and concerns of patients in the palliative phase with limited health literacy may be of a different order or more prevalent than in other patients in the curative phase.’

Besides a problem with phrasing (‘different order’?) I wonder whether this sentence is justified. We see a mixture of two factors mentioned: low versus high or average health literacy, and curative versus palliative phase of a disease. I think that these two factors cannot be disentangled within this design – there are no data on patients in a similar phase of their disease being non-low in health literacy.

Here your concern is also shared by reviewer #1. Below the reaction on this point. 

There are indeed two variables when it comes to comparing the findings with other studies; the palliative phase and health literacy of patients. The population of this study consists of lung cancer and COPD patients with limited health literacy in the palliative phase. Especially patients with limited health literacy is a specific population that is mostly not included in research. In the manuscript we have compared the results of our study to those of Del Piccolo et al.[1], Finset, Heyn and Ruland[2] and Oguchi et al.[3], which all included cancer patients in their study population. However, none of these studies did specifically include patients in the palliative phase nor selected for health literacy. To the best of our knowledge there are no studies into affective communication in patients with limited health literacy using the VR-CoDES. 

To address your remark to make a comparison between patients in the curative or palliative setting and between low and high literate patients, we have expanded the discussion by including the article of Amundsen et al.[4] in which it is shown that the setting (either curative or palliative) did not influence the number of cues and concerns expressed by patients at the Oncology Outpatient Clinic at the University Hospital of North Norway. Again, this study did not include LHL patients. This leads us to speculate that the increased number of cues and concerns expressed by this specific patient population is indeed related to the health literacy of the patients, rather than to the specific phase of the disease, either curative or palliative.

 ‘This is relevant because the patients in this study expressed cues and concerns indicating distressing misunderstandings.’ Did I miss this point? Where do the authors present data on the nature of the cues and concerns that would indicate that they are related to misunderstandings? Is this based upon the assumption that high health literacy is effective against misunderstandings? Then, this connection should be made explicit. Basically, I am not so sure, whether this is the case. Couldn’t it be that hcps adjust their wording and the complexity of information to their perception of patients’ health literacy in such a way that ‘smarter’ patients are getting more information thereby offsetting the benefit of a higher health literacy? The very next sentence in the discussion paper would rather strengthen my skeptical comment: patients refer to their fear of suffocation and that’s not a misunderstanding but harsh reality.

Although the data indeed indicates that patients express cues and concerns related to misunderstandings, the authors agreed that there is not enough data to support this statement but rather focus on the diagnose-specific cues and concerns (for example Quote 6). It is tempting to suggest what could be the reason for the increased amount of cues and concerns expressed by the study population compared to the general population (those with sufficient health literacy), but we agree with earlier remarks that caution should be exercised when it comes to formulation conclusions based on an explorative study. Therefore the authors choose to focus on the evidently higher amount of cues and concerns in this specific study, rather than looking at the nature of the cues and concerns. Further qualitative research has to be done into the exact nature of the cues and concerns. 

In the manuscript we addressed this comment by reviewing the section ‘added value of this study’ under Discussion. Here we stated that: 

‘Nevertheless, given that we know that patients with limited health literacy are more likely to experience difficulties in provider-patient communication,[8] it seems possible that these difficulties are reflected in the amount of expressed cues or concerns. In addition, the relatively high prevalence of cues and concerns about the fear of suffocation (Quote 6) would seem to be specific to the patient population (lung patients). Further studies, which take these difference settings and patient populations into account, will need to be undertaken.’ 

References

1. Del Piccolo L, Mazzi MA, Mascanzoni A, Lonardi M, De Felice M, Danzi OP, et al. Factors related to the expression of emotions by early-stage breast cancer patients. Patient education and counseling. 2019;102(10):1767-73.

2. Finset A, Heyn L, Ruland C. Patterns in clinicians’ responses to patient emotion in cancer care. Patient Education and Counseling. 2013;93(1):80-5.

3. Oguchi M, Jansen J, Butow P, Colagiuri B, Divine R, Dhillon H. Measuring the impact of nurse cue-response behaviour on cancer patients’ emotional cues. Patient Education and Counseling. 2011;82(2):163-8.

4. Amundsen A, Nordøy T, Lingen KE, Sørlie T, Bergvik S. Is patient behavior during consultation associated with shared decision-making? A study of patients’ questions, cues and concerns in relation to observed shared decision-making in a cancer outpatient clinic. Patient education and counseling. 2018;101(3):399-405.

5. WHO. WHO definition of palliative care, 2002. Available from: https://www.who.int/health-topics/palliative-care. 

6. Sepúlveda C, Marlin A, Yoshida T, Ullrich A. Palliative care: the World Health Organization's global perspective. Journal of pain and symptom management. 2002;24(2):91-6.

7. Barrett SE, Puryear JS. Health literacy: improving quality of care in primary care settings. Journal of Health Care for the Poor and Underserved. 2006;17(4):690-7.

8. Oosterveld M, Noordman J, Rademakers J. Kennisvraag. Samen beslissen in de spreekkamer. Ervaringen en behoeften van mensen met beperkte gezondheidsvaardigheden. 2019.

---

## [Decision Letter · Decision Letter 1]

20 Jan 2022

Affective communication with patients with limited health literacy in the palliative phase of COPD or lung cancer: analysis of video-recorded consultations in outpatient care

PONE-D-21-01177R1

Dear Dr. Otte,

We’re pleased to inform you that your manuscript has been judged scientifically suitable for publication and will be formally accepted for publication once it meets all outstanding technical requirements.

Kind regards,

Barbara Schouten

Academic Editor

PLOS ONE

**Comments to the Author**

1. If the authors have adequately addressed your comments raised in a previous round of review and you feel that this manuscript is now acceptable for publication, you may indicate that here to bypass the “Comments to the Author” section, enter your conflict of interest statement in the “Confidential to Editor” section, and submit your "Accept" recommendation.

Reviewer #1: All comments have been addressed

Reviewer #2: All comments have been addressed

2. Is the manuscript technically sound, and do the data support the conclusions?

Reviewer #1: Yes

Reviewer #2: Yes

3. Has the statistical analysis been performed appropriately and rigorously? 

Reviewer #1: Yes

Reviewer #2: Yes

4. Have the authors made all data underlying the findings in their manuscript fully available?

Reviewer #1: Yes

Reviewer #2: Yes

5. Is the manuscript presented in an intelligible fashion and written in standard English?

Reviewer #1: Yes

Reviewer #2: Yes

6. Review Comments to the Author

Reviewer #1: My comments to the first submission of this manuscript have been satisfactorily handled. No further comments.

Reviewer #2: Thanks for carefully addressing the two reviewers' comments. It's reassuring to read that both had similar comments

7. PLOS authors have the option to publish the peer review history of their article (what does this mean?). If published, this will include your full peer review and any attached files.

Reviewer #1: No

Reviewer #2: **Yes: **Wolf Langewitz

---

## [Editor Report · Acceptance letter]

24 Jan 2022

PONE-D-21-01177R1 

Affective communication with patients with limited health literacy in the palliative phase of COPD or lung cancer: analysis of video-recorded consultations in outpatient care 

Dear Dr. Otte:

I'm pleased to inform you that your manuscript has been deemed suitable for publication in PLOS ONE. Congratulations! Your manuscript is now with our production department. 

Kind regards, 

on behalf of

Dr. Barbara Schouten 

Academic Editor

PLOS ONE